# Retrieval-Augmented Generation as In-Context Optimization: A Gradient Descent Perspective

## Abstract

Currently, it remains unclear whether in-context learning (ICL) can serve as an alternative mechanism for retrieval-augmented generation (RAG), and its underlying operation is still poorly understood and largely intuitive. In this paper, we propose that trained Transformers can be viewed as performing retrieval-augmented generation through gradient descent. We start by proving a weight construction and showing the equivalence of data transformations induced by linear self-attention-based Transformer and RAG training on a regression loss. Motivated by this construction, we empirically demonstrate that, when trained on simple regression tasks, self-attention-only Transformers exhibit strong similarity to RAG models trained via gradient descent. This allows us, at least within the scope of regression problems, to gain a mechanistic understanding of how in-context learning can be leveraged to optimize RAG. Moreover, we observe that the distribution of the data critically affects the generalizability of the learned models in the non-linear setting, so we propose strategies to enhance the robustness of in-context learning (ICL) against distributional variability encountered in practice. Among these, we explore normalization techniques as one representative approach, showing that they can effectively improve both stability during training and generalization across domains.

## 1 Introduction

Large Language Models (LLMs) have achieved remarkable progress across a wide range of natural language processing (NLP) tasks, including link prediction, question answering, text classification, and text generation (Li & Ji, 2022; Achiam et al., 2023; Li et al., 2024b; Guo et al., 2025). However, it is challenging for LLMs to incorporate newly emerging knowledge beyond their static pre-training data. To address this, retrieval-augmented generation (RAG) has emerged as an effective solution (Lewis et al., 2020; Li et al., 2025), enabling LLMs to retrieve relevant information from external corpora, thereby enhancing performance on downstream tasks.

There are three approaches to investigating the retrieval-augmented generation (RAG) capabilities of attention-based generative models. The first is that zero-shot learning (ZSL) methods (Huang et al., 2024) enable the retriever to identify relevant documents and supply them, together with the input sentences, to the predictor model, where both the retriever and predictor remain frozen. While this approach is computationally efficient, it struggles with personalization, as it fails to incorporate specific user behaviors or preferences. For the second, fine-tuning methods (Li et al., 2024e; Zhang et al., 2025a) adapt the trainable retriever and predictor, while this approach can improve performance, it requires substantial computational resources for retraining. Recently, in-context learning (ICL) has been leveraged in retrievers and generators by conditioning a number of input–output examples (Li et al., 2024d), thereby eliminating the need for additional parameter updates or task-specific training. Compared with fine-tuning, this approach can substantially reduce computational overhead and time consumption. Despite the efficiency advantages of applying ICL in RAG, no prior work has systematically investigated, either theoretically or empirically, whether ICL can achieve accuracy comparable to jointly fine-tuning the retriever and generator in RAG. In this work, we address this gap by analyzing whether ICL can serve as an alternative to explicit training in RAG systems, with the goal of achieving a better balance between efficiency and accuracy.

To address this objective, we identify two major gaps: 1) A growing body of work has investigated how Transformers can approximate fine-tuning through gradient descent, viewing it as a mechanism for ICL. For instance, studies such as (Von Oswald et al., 2023; Gatmiry et al., 2024; Mahankali et al., 2023) demonstrate that models can learn to perform preconditioned gradient descent on input examples, with explicit weights converging toward the global minimum. More recent research (Shen et al., 2025; Ren & Liu, 2024) examines ICL from the perspective of kernel functions, by reformulating the attention layer as a linear kernel mapping. These studies have achieved substantial theoretical progress in characterizing the mechanisms underlying ICL and have shown that, despite being confined to linear regression settings, they can nonetheless capture key behaviors observed in nonlinear architectures, thereby serving as a convincing proxy for understanding ICL in nonlinear. Nevertheless, these studies have primarily considered ICL as generators, and, to the best of our knowledge, no theoretical analysis has explored ICL in the joint setting of trainable retrievers and generators. 2) Current studies mainly benchmark the performance of different trainable RAG models, for example, Li et al. (2024c) evaluates three retrievers in combination with four different generators. In contrast, other approaches focus on evaluating the effectiveness of ICL (Wang et al., 2024; Mosbach et al., 2023). For example, Mosbach et al. (2023) evaluates the generalization ability of in-context learning on in-domain and out-of-domain knowledge. However, no prior work has conducted a systematic benchmarking or comparative analysis of the performance differences between ICL and trainable RAG under the unified experimental setting (e.g., using the same dataset).

So, in this paper, we begin by establishing the expressiveness of ICL, showing that trained Transformers can act as a learning algorithm capable of performing both retrieval and generation. Specifically, We demonstrate that the Transformer (1) formulates a loss function for RAG that depends on the input question and the documents to be retrieved, and (2) updates its parameters by learning from the gradient of this loss using the constructed weight values. We provide the theoretical proof for a Transformer with a single self-attention layer under a linear architecture, and then extend the analysis to deeper architectures. Finally, we evaluate the expressiveness of ICL as RAG models on four widely used regression tasks in the nonlinear setting, and provide a systematic comparative analysis of the performance between ICL and trainable RAG. Our findings reveal that their generalizability is constrained by the complexity and variability of data distributions. Building on this insight, we identify the key challenges posed by distributional shifts and propose normalization techniques to enhance the robustness of ICL in RAG systems. Our contributions are summarized as follows:

- By explicitly constructing the weight matrices, we demonstrate that a linear self-attention layer performs updates in the form of a weighted sum over input features. This operation is mathematically equivalent to training a simplified RAG system for joint document selection and output prediction under a mean squared error objective. Furthermore, we show that the composition of multiple self-attention layers can iteratively approximate curvature correction, making the optimization trajectory closely approximate that of conventional RAG training in terms of convergence properties.

- When optimized on linear regression datasets, self-attention–only Transformers with constructed weights can emulate the gradient descent training process of RAG, both on in-distribution and out-of-distribution validation tasks.

- We extend the analysis of ICL and RAG beyond linear settings and evaluate ICL on four widely used regression tasks under the non-linear settings. Our results indicate that its generalizability is substantially limited when the underlying data distribution deviates from idealized assumptions. To address this limitation, we propose the use of normalization techniques to mitigate the distributional gaps and enhance robustness.

## 2 SELF-ATTENTION CAN EMULATE THE GRADIENT-BASED TRAINING OF RAG ON A LINEAR REGRESSION TASK

In this section, we first introduce the linear self-attention based Transformer, followed by the definition of the RAG models with two types of retrievers: a linear projection retriever and a dot product retriever, building on prior work. We then establish the connection between the training RAG and self-attention by the data transformation. Finally, we present a proposition demonstrating the equivalence between the training process of RAG and the linear self-attention model.

## 2.1 SELF-ATTENTION

To illustrate the connection between in-context learning and the training progress of RAG by gradient descent, we begin with a multi-head self-attention (SA) block parameterized by $\theta$. Given a sequence of tokens $\{e_1, \ldots, e_N\}$, the update rule for an element $e_j$ can be expressed as

$$e_j \leftarrow e_j + \text{SA}_\theta(j, \{e_1, \ldots, e_N\}) = e_j + \sum_h P_h V_h \, \text{softmax}(K_h^\top q_{h,j}), \tag{1}$$

where $P_h$, $V_h$, and $K_h$ denote the projection, value, and key matrices for the $h$-th head, and $q_{h,j}$ is the query corresponding to position $j$. The value matrix is defined as $V_h = [v_{h,1}, \ldots, v_{h,N}]$ with $v_{h,i} = W_{h,V} e_i$, while the key matrix is $K_h = [k_{h,1}, \ldots, k_{h,N}]$ with $k_{h,i} = W_{h,K} e_i$. Similarly, the query vector is computed as $q_{h,j} = W_{h,Q} e_j$. Altogether, the parameters of the SA block are given by $\theta = \{P_h, W_{h,V}, W_{h,K}, W_{h,Q}\}_{h=1}^H$, collecting all projection matrices across the $H$ attention heads. We introduce a key modification to the standard self-attention mechanism by removing the softmax operation and bias terms. This architecture was also used in previous work (Von Oswald et al., 2023; Vladymyrov et al., 2024). Based on this motivation, we derive the *linear self-attention* (LSA) update rule:

$$e_j \leftarrow e_j + \text{LSA}_\theta(j, \{e_1, \ldots, e_N\}) = e_j + \sum_h P_h V_h K_h^\top q_{h,j}. \tag{2}$$

## 2.2 RETRIEVAL ARGUMENTED GENERATION MODEL

Building upon the reference linear model $y(x) = Wx$ of in-context learning introduced by Von Oswald et al. (2023), we generalize the formulation to the RAG framework, where we incorporate two distinct retrieval mechanisms, RAG with linear projection retriever and RAG with dot product retriever.

**RAG with Linear Projection Retriever**   Inspired by the marginalization mechanism in RAG-Sequence (Lewis et al., 2020), where the generator is conditioned on both the query and the retrieved documents. We incorporate an additional projection matrix $W_d$ in the reference linear model that operates on a document embedding $D = (d_1, d_2, d_3, \ldots, d_n)^\top$, where each $d_i$ corresponds to the embedding of a retrieved document. The projection matrix $W_d$ maps each retrieved document embedding into a $k$-dimensional representation, which is then combined with the input representation through $W_1$.

The resulting model can be expressed as: $y = (W_1 \quad W_2) \begin{pmatrix} x_q \\ W_d D \end{pmatrix}, y = W_1 x_q + \sum_{i=1}^k W_z d_i$, where

$W_z \triangleq W_2 W_d$, $x^q \in \mathbb{R}^{d_q}$, $D \in \mathbb{R}^{k \times d_d}$, $W_d \in \mathbb{R}^{d_m \times k}$, $W_1 \in \mathbb{R}^{d_y \times d_q}$, $W_2 \in \mathbb{R}^{d_y \times d_m}$. $W_z$ is regarded as the retriever, where the retrieved documents together with the input query $x_q$ are used to predict the final output $y$. This formulation highlights how retrieval-based features, encoded through $W_d$, can be integrated with the original input representation $x_1$ via linear projection. Let the training dataset be $\mathcal{D} = \{(x_i^q, x_i^d, y_i)\}_{i=1}^N$, with inputs $x_i^q \in \mathbb{R}^{N_x}$ and corresponding labels $y_i \in \mathbb{R}^{N_y}$.

**RAG with Dot Product Retriever**   For the dot product retriever (Karpukhin et al., 2020), given a query input $x$ and a set of candidate documents $\{d_i\}_{i=1}^n$, The linear encoder, represented by the weight matrix $W_e$ is used to compute the embeddings of the query and candidate documents: $x = W_e(x)$, $d_i = W_e(d_i)$, to align our formulation with the reference linear model, we adopt this design choice following prior work (Xie et al., 2017). Subsequently, the input query, together with the scored documents, is used to generate the final prediction. In the linear case, the attention weight $\alpha_i$ assigned to each document $d_i$ is determined by the similarity score between the query and document embeddings: $\alpha_i = (W_e x_q)^\top (W_e d_i)$. Accordingly, the prediction function is defined as,

$$y = (W_q, \ W_z) \begin{pmatrix} x_q \\ \sum_{i=1}^n (W_e x_q)^\top (W_e d_i) \, d_i \end{pmatrix} = W_q x_q + W_z \sum_{i=1}^n (W_e x_q)^\top (W_e d_i) \, d_i \tag{3}$$

For notational convenience, we define $M = W_e^\top W_e$ in the attention weight $\alpha_i$, yielding $\alpha_i = x_q^\top M d_i$. Moreover, in equation 3, note that $\sum_{i=1}^n (x_q^\top M d_i) d_i = \sum_{i=1}^n d_i d_i^\top M^\top x_q$, let $C =$

$\sum_{i=1}^{n} d_i d_i^{\top}$, so the above expression can be simplified as $\sum_{i=1}^{n} (x_q^{\top} M d_i) d_i = C M^{\top} x_q$. Setting $W = W_z C M^{\top}$, we obtain $y = W_q x_q + W_z C M^{\top} x_q$, which leads to the compact formulation below.

$$y = W_q x_q + W x_q, \quad \text{where } W = W_z \left( \sum_{i=1}^{n} d_i d_i^{\top} \right) M^{\top}. \tag{4}$$

where $x_q \in \mathbb{R}^{d_d}, \quad d_i \in \mathbb{R}^{d_d}, \quad W_e \in \mathbb{R}^{r \times d_d}, \quad W_q \in \mathbb{R}^{d_y \times d_d}, \quad W_z \in \mathbb{R}^{d_y \times d_d}, M = W_e^{\top} W_e \in \mathbb{R}^{d_d \times d_d}, \quad C = \sum_{i=1}^{n} d_i d_i^{\top} \in \mathbb{R}^{d_d \times d_d}, \quad y \in \mathbb{R}^{d_y}.$

**United RAG with different retrievers** In this section, we unify the above linear RAG formulation as $y = W_1 x_1 + W_2 x_2$, where $W_1 \in \mathbb{R}^{N_y \times N_{x1}}, \quad W_2 \in \mathbb{R}^{N_y \times N_{x2}}$, the matrix $W_1$ functions as a feature selection and weighting operator on the query embedding, allowing the model to emphasize task-relevant components while suppressing irrelevant ones. In the linear projection retriever, $x_1 = x_q, x_2 = D, W_2 \triangleq W_1 W_d$. In the dot product retriever, $x_1 = x_2 = x_q, W_2 = W_z \left( \sum_{i=1}^{n} d_i d_i^{\top} \right) M^{\top}$.

### 2.3 Optimization Objective for Unified RAG

Building on the unified RAG formulation, we now formalize the learning objective by minimizing the squared error loss:

$$L(W_1, W_2) = \frac{1}{2N} \sum_{i=1}^{N} \| W_1 x_1^i + W_2 x_2^i - y_i \|^2 \tag{5}$$

One step of gradient descent on $L$ with learning rate $\eta$ yields the following update, where the weight change is defined as:

$$\Delta W_1 = -\eta \nabla_{W_1} L(W_1) = -\frac{\eta}{N} \sum_{i=1}^{N} \left( W_1 x_1^i + W_2 x_2^i - y_i \right) \left( x_1^i \right)^{\top}$$

$$\tag{6}$$

$$\Delta W_2 = -\eta \nabla_{W_2} L(W_2) = -\frac{\eta}{N} \sum_{i=1}^{N} \left( W_1 x_1^i + W_2 \, x_2^i - y_i \right) \left( x_2^i \right)^{\top}$$

In the transformed targets $y_i - \Delta y_i$, $\Delta y_i = \Delta y_i^1 + \Delta y_i^2 = \Delta W_1 x_1 + \Delta W_2 x_2$. Thus, the outcome of a gradient step can be interpreted as an update to the regression loss. The update of the target $y$ depends on the input data $x$ and $\Delta W$.

### 2.4 Training linear RAG with Gradient Descent Is Equivalent to a Linear Self-Attention Layer

We reinterpret the learning RAG in the linear model as a transformation applied directly to the data. This perspective allows us to draw a connection between RAG and linear self-attention. Specifically, we describe a construction in which learning occurs simultaneously for all tokens, including the test token, through updates induced by a linear self-attention layer. In this view, the response to a query (test) token is transformed from its initial state $W_1 x_{\text{test}}^1, W_2 x_{\text{test}}^2$, to the post-update prediction $\hat{y} = (W_1 + \Delta W_1) x_{\text{test}}^1 + (W_2 + \Delta W_2) x_{\text{test}}^2$ which corresponds to the result obtained after a single step of gradient descent.

**Lemma** (Equivalence). *Given a 1-head linear attention layer and tokens $e_j = (x_1^j, x_2^j, y^j)$ for $j = 1, \ldots, N$, we can construct special key, query, and value matrices $W_K, W_Q, W_V$, together with a projection matrix $P$, such that a Transformer update on each token $e_j$ is equivalent to the training progress of the above RAG optimization.*

*Proof.* $e_j \leftarrow (x_1^j, x_2^j, y^j) + (0, -\Delta W_1 x_1^j, -\Delta W_2 x_2^j) = (x_1^j, x_2^j, y^j) + P V K^{\top} q_j$, the target is updated according to the weight construction method described below.

$$\begin{pmatrix} x_1^j \\ x_2^j \\ y^j \end{pmatrix} \leftarrow \begin{pmatrix} x_1^j \\ x_2^j \\ y^j \end{pmatrix} + \begin{pmatrix} 0 \\ 0 \\ -(\Delta W_1 x_1 + \Delta W_2 x_2) \end{pmatrix} \tag{7}$$

$$\begin{pmatrix} x_1^j \\ x_2^j \\ y^j \end{pmatrix} \leftarrow \begin{pmatrix} x_1^j \\ x_2^j \\ y^j \end{pmatrix} + \frac{\eta}{N} \sum_{i=1}^{N} \left( \underbrace{\begin{pmatrix} 0 & 0 & 0 \\ 0 & 0 & 0 \\ W_1 & W_2 & -I_y \end{pmatrix}}_{W_V} \begin{pmatrix} x_1^i \\ x_2^i \\ y^i \end{pmatrix} \right) \otimes \left( \underbrace{\begin{pmatrix} I_x & 0 & 0 \\ 0 & I_x & 0 \\ 0 & 0 & 0 \end{pmatrix}}_{W_K} \begin{pmatrix} x_1^i \\ x_2^i \\ y^i \end{pmatrix} \right)^{\top} \left( \underbrace{\begin{pmatrix} I_x & 0 & 0 \\ 0 & I_x & 0 \\ 0 & 0 & 0 \end{pmatrix}}_{W_Q} \begin{pmatrix} x_1^j \\ x_2^j \\ y^j \end{pmatrix} \right) \tag{8}$$

The right side of equation 7 is identical to the right side of equation 8, implying that the RAG optimization is equivalent to the Transformer update. Where we are provided with $N$ context tokens together with an additional query token, indexed by $N + 1$. Each context token can be written as $e^i = (x_1^i, x_2^i, y^i)$, representing one of the $N$ training pairs. The $(N + 1)$-th token is given by $e_j = (x_1^j, x_2^j, y^j)$, $y^j = 0$. The model is expected to predict the updated $y^j$ value. Please refer to Appendix E for the detailed mathematical proof. □

## 3 EXPRESSIVENESS OF ICL AS GRADIENT DESCENT ON RETRIEVAL-AUGMENTED GENERATION

In this section, we experimentally investigate whether attention-based models are capable of realizing gradient-based RAG during their forward computation. We incrementally extend our analysis, beginning with a single linear self-attention layer and extending to multi-layer nonlinear architectures.

**Transformer Pre-Training** We define each token by concatenating input data, the auxiliary representation, and the target, $e_i = (x_i, z_i, y_i)$, $1 \le i \le N$, $N$ is the size of the training data for each task $\tau$. $z_i$ is the document embedding or $x_i$. The training objective is to minimize the expected squared prediction error across tasks. Formally,

$$\mathcal{L}(\theta) = \frac{1}{B} \sum_{\tau=1}^{B} \left\| \hat{y}_\theta \left( \{e_{\tau,i}\}_{i=1}^{N}, e_{\tau,N+1} \right) - y_{\tau,N+1} \right\|^2 \tag{9}$$

where, for each task $\tau$, $e_{\tau,N+1} = (x_{\text{test}}, z_{\text{test}}, 0)$, $y_{\tau,N+1}$ is the target of $e_{N+1}$, that is, $y_{\tau,N+1} = (x_{\text{test}}, z_{\text{test}}, y_{target})$. This objective is optimized using minibatch stochastic gradient descent. At each iteration, we sample a batch of training tasks and update the parameters $\theta$ by taking a gradient step on the empirical loss. We denote the optimal parameters after training by $\theta^*$.

Following Garg et al. (2022); Von Oswald et al. (2023), we generate data for each task using a teacher model with parameters $W_\tau \sim \mathcal{N}(0, I)$. For each task, we sample inputs $x_{\tau,i} \sim U(-1, 1)^{n_I}$ and construct targets through the task-specific teacher, $y_{\tau,i} = W_\tau^1 x_{\tau,i}^1 + W_\tau^2 x_{\tau,i}^2$. In our experiments, for the synthetic data, we set $N = n_I = 10$, $n_I$ is the feather size, the output dimension is set to 1. In our experiments, we investigate the effect of varying the document size $k \in \{2, 5, 10, 25\}$.

Unlike RAG with the linear projection retriever, where the document is explicitly included in the input as $e_i = (x_i, \mathcal{D}, y_i)$ and the model can acquire the ability to select relevant documents during pre-training, in the dot-product retriever setting, the document cannot be directly concatenated into the input, instead, we inject the document knowledge into the key and value matrices of the Transformer: $K = \begin{bmatrix} K_{\text{ctx}} \\ h_d \end{bmatrix}$, $V = \begin{bmatrix} V_{\text{ctx}} \\ h_d \end{bmatrix}$, where $K_{\text{ctx}}$ and $V_{\text{ctx}}$ denote the contextual key and value representations. Given a document set $\mathcal{D} = \{d_1, d_2, \ldots, d_n\}$, the mapping function $f(.)$ projects $\mathcal{D}$ into the same dimensional space as the input data $x$ in a batch size $B$, yielding $h_d = f(\mathcal{D}) \in \mathbb{R}^{B \times \dim(x)}$.

**Prediction using a trained transformer** When given a new task $\tau$ that is not included in the training set, in the testing progress, the query token is constructed by concatenating the test input, auxiliary representation, a zero vector for the target, yielding $e_{N+1}^* = (x_{\text{test}}^*, z_{\text{test}}^*, 0)$. The prediction of the attention-based model is given by $\hat{y}_{\theta^*}(x_{\text{test}}) = \hat{y}_{\theta^*} \left( \{e_{\tau,1}^*, \ldots, e_{\tau,N}^*\}, e_{\tau,N+1}^* \right)$ which depends on all tokens and the model parameters $\theta^*$, learned during the training process. The output is read from the $y$-component of the updated $(N + 1)$-th token. So, the output is $\hat{y}_{\theta^*}(x_{\text{test}})$.

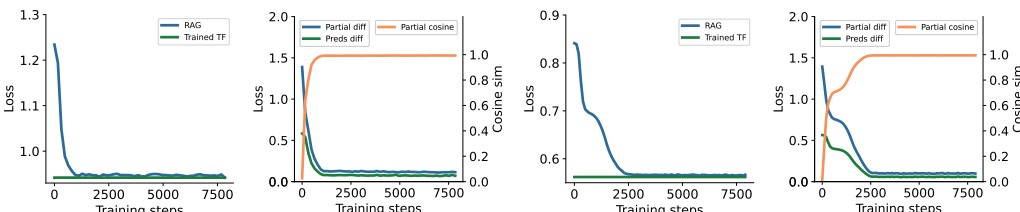

Figure 1: Comparison of one-step training of the linear RAG model with a trained single linear self-attention (LSA) layer. *Outer left*: The loss of the trained LSA layer matches that of the RAG model trained via gradient descent. *Center left*: After training, the RAG model and the LSA-based model exhibit near-perfect alignment, as measured by cosine similarity and $\ell_2$ distance between both models and their predictions. *Center right and Outer right* Results for the linear projection layer, whereas *Outer left and Center left* correspond to the dot-product layer, both evaluated under the same evaluation metrics.

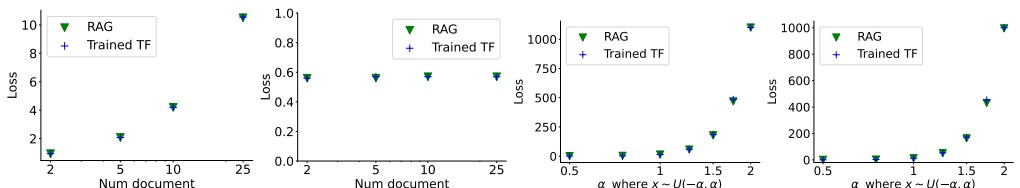

Figure 2: Comparing one step of RAG training with a trained single self-attention layer across different document counts and scaling factors, we observe that the trained LSA layer, gradient descent, and their interpolation yield nearly identical losses (in log scale) even when the test data distribution differs from training, i.e., at a scale of 1. Outer left: Loss comparison between RAG and the trained Transformer for varying document numbers under a linear projection retriever. Center left: Loss comparison between RAG and the trained Transformer for varying document numbers under a dot product retriever.

**Prediction using RAG training** In equation 8, by defining distinct weight matrices $W_1$ and $W_2$ within $W_V$ and combining them with the projection matrices $W_K$ and $W_Q$, we construct weighted interaction terms over the context input $\{e^*_{\tau,1}, \ldots, e^*_{\tau,N}\}$ and test point $e^*_{\tau,N+1}$. This update rule yields a output $\hat{y}_{\theta,\mathrm{rag}}(x_{\mathrm{test}})$, that is determined by the joint effect of these weight matrices, thereby enabling controlled variation in the model output. For the RAG framework with a linear projection retriever, we obtain the controlled model output $\hat{y}_{\theta,\mathrm{rag}}(x_{\mathrm{test}})$ by initializing different weight matrices $W_1$ and $W_2$ within the projection layers $W_K, W_Q$, and $W_V$ in equation 8. Following Von Oswald et al. (2023), we initialize the weight matrices $W_1$ and $W_2$ as zero matrices. In the RAG with dot product retriever, $W_2 = W_z \left( \sum_{i=1}^n d_i d_i^\top \right) M^\top$, we initialize all parameters from a zero-mean Gaussian distribution with variance $\sigma^2$. Specifically, $W_z \in \mathbb{R}^{d_y \times d_d}$, $M \in \mathbb{R}^{d_d \times d_d}$, are sampled independently as $W_z, M \sim \mathcal{N}(0, \sigma^2)$, $C \sim \mathcal{U}\left(-\frac{1}{2}, \frac{1}{2}\right)^{k \times d_d}$, $k$ is the number of documents. The document covariance matrix is then constructed as $\sum_{i=1}^k d_i d_i^\top \in \mathbb{R}^{d_d \times d_d}$ and we define $W_2 = W_z C M^\top \in \mathbb{R}^{d_y \times d_q}$. For this control model, we determine the optimal learning rate $\eta$ by minimizing $\mathcal{L}(\eta)$ over a training set of $10^4$ tasks using line search.

**Evaluation** More concretely, to compare trained and constructed LSA layers, Same as Von Oswald et al. (2023) we sample $T_{\mathrm{val}} = 10^4$ validation tasks and record the following quantities, averaged over validation tasks: 1)The difference in predictions, measured with the L2 norm, $\|\hat{y}_\theta(x_{\tau,\mathrm{test}}) - \hat{y}_{\theta,\mathrm{rag}}(x_{\tau,\mathrm{test}})\|$ 2)The cosine similarity between the sensitivities $\frac{\partial \hat{y}_{\theta,\mathrm{rag}}(x_{\tau,\mathrm{test}})}{\partial x_{\mathrm{test}}}$ and $\frac{\partial \hat{y}_\theta(x_{\tau,\mathrm{test}})}{\partial x_{\mathrm{test}}}$. 3) Their difference, according to the L2 norm, $\left\| \frac{\partial \hat{y}_{\theta,\mathrm{rag}}(x_{\tau,\mathrm{test}})}{\partial x_{\mathrm{test}}} - \frac{\partial \hat{y}_\theta(x_{\tau,\mathrm{test}})}{\partial x_{\mathrm{test}}} \right\|$.

**Results** 1) We show the results of these comparisons in Figure 1. We find an excellent agreement between the RAG with two types of retrievers and the trained self-attention layer. Beyond the

in-distribution setting, we further analyze the behavior of RAG training under out-of-distribution (OOD) conditions as well as with different numbers of retrieved documents. Specifically, we evaluate whether the trained self-attention layer continues to align with RAG when the test distribution deviates from training (i.e., out of scale), and when the number of retrieved documents varies. These analyses demonstrate the robustness of the correspondence: the trained Transformer, RAG, and their interpolation exhibit nearly identical loss trends even in OOD scenarios and across varying retrieval sizes. 2) To evaluate whether the in-context learner captures a generalizable update rule, we examine the training behavior of RAG with two types of retrievers and a trained linear self-attention (LSA) layer, under a setting where the testing data distribution differs from that of the training data. Specifically, we measure the loss under sampling the input query from $U(-\alpha, \alpha)^{n_I}$ with varying $\alpha$, in this work, we set the $\alpha = 0.5, 1, 1.5, 2$ for the testing data. During training, we fix $\alpha = 1$, while at test time we alter its value to probe robustness. In both cases, the single-layer transformer closely matches the RAG training even outside of the training regime, as illustrated in Figure 2. 3) As illustrated in Figure 2, We compare the loss of RAG training with gradient descent and the trained Transformer under varying retrieval sizes. in this work, we set the number of document are $n = 2, 5, 10, 25$ With a linear projection retriever, the loss increases significantly as the number of retrieved documents grows, although the trained Transformer continues to follow gradient descent closely. In contrast, with a dense retriever, where documents are directly embedded into the weight matrix $W$, the loss remains largely unaffected by the retrieval size. This not only demonstrates a closer alignment between RAG and the Transformer, but also highlights that the dot product retriever provides faster computation than the linear projection retriever, as no additional projection step is required.

**Multiple-step training of RAG**   In this section, we explore the deep linear self-attention Transformers. The framework established in our defined proposition naturally extends to $K$ stacked layers. In this setting, the final prediction is again determined from the $y$-coordinate of the last test token. Specifically, after $K$ updates we obtain

$$y_{N+1} + \sum_{k=1}^{K} \Delta y_{k, N+1} = y_{N+1} + \sum_{k=1}^{K} (W_k^1 x_{N+1}^1 + W_k^2 x_{N+1}^2), \tag{10}$$

where $y_{k, N+1}$ denotes the value of the test token at layer $k$, and $\Delta y_{k, N+1}$ represents the increment in the $y$-coordinate after the $k$-th self-attention update. The term $W_k$ corresponds to the implicit change in the underlying linear model parameters $W$ induced by the $k$-th attention step. To investigate the impact of increasing model depth, we consider the simplest extension beyond a single self-attention layer: a two-layer LSA model with shared parameters. In this setting, the same layer is applied multiple times, effectively reusing identical weights across iterations. This design can be viewed as learning an iterative procedure, where the model refines its representation by repeatedly applying the same transformation. In addition, we configure retrieval pools of different sizes for the two retriever variants, enabling a direct comparison of their behavior under varying retrieval capacities.

As shown in Figure 3, we present the experimental results of RAG with a dot-product retriever. The loss differences between the trained Transformer and RAG remain closely aligned across different document numbers, and the prediction differences also converge to similar values. We further observe that the number of retrieved documents affects the degree of equivalence: with two layers, the prediction difference at Docs=2 is smaller than at Docs=25. This gap, however, becomes less pronounced as the model depth increases, for instance, at five layers the discrepancy is considerably reduced. For further analysis of RAG with a linear-projection retriever, please refer to Appendix B.

## 4   NORMALIZATION FOR MITIGATING DISTRIBUTIONAL SHIFTS IN ICL

It is not reasonable to assume that training a RAG model is equivalent to in-context learning (ICL) in linear Transformers when restricted to linear models and synthetic datasets. So, we extend the setting by introducing MLP layers after the input embedding in the Transformer, thereby incorporating nonlinearity into the generator. In this section, we mainly focus on RAG with a dot-product retriever and analyze the performance differences between ICL and trainable RAG models.

For our empirical evaluation, we employed four publicly available real-world datasets for regression tasks: California Housing, Bike Sharing, Wine Quality, and Predict Calorie Expenditure (sourced

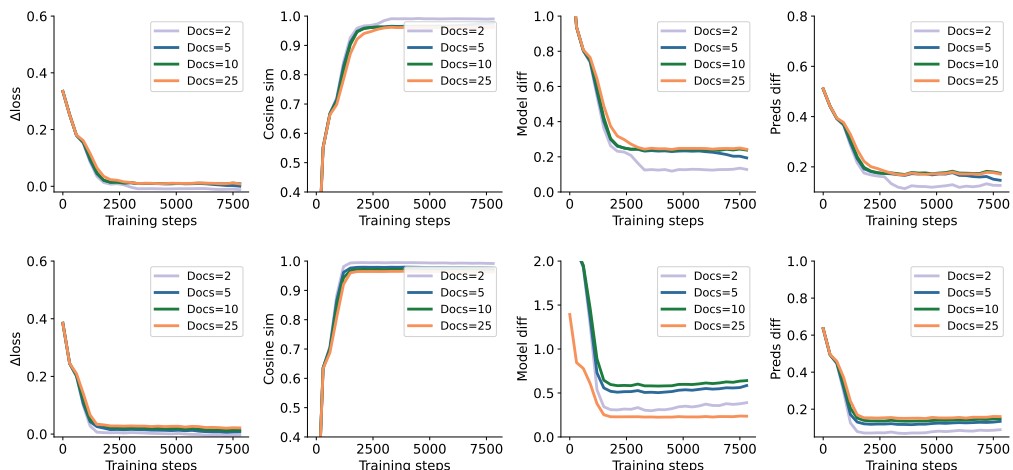

Figure 3: Comparison across layers with varying numbers of documents for the dot product retriever. The first row corresponds to models with 2 layers, and the second row corresponds to models with 5 layers. Each column reports a different evaluation metric: (a) loss difference between the trained Transformer and RAG, (b) cosine similarity, (c) model difference, and (d) prediction difference.

from Kaggle). A detailed description of each dataset is provided in Appendix F. To examine the effect of feature scaling, we applied four normalization techniques: Z-score normalization (Bishop & Nasrabadi, 2006), Min–Max normalization (Bishop & Nasrabadi, 2006), Rank-based normalization (Conover, 1999), and Tanh normalization (Maaten & Hinton, 2008). In our experimental setup, the training set was used as the retrieval corpus. The retrieval corpus was normalized using Z-score normalization, while the input data were separately normalized under each of the four normalization schemes for comparison across retrievers.

As shown in Figure 4. In the Bike Sharing dataset, applying Min–Max normalization yields performance for the trained Transformer that closely matches that of RAG. In real-world datasets, feature distributions are often bounded and non-Gaussian, making Z-score normalization less effective. Min–Max scaling, by contrast, uniformly maps all features into the [0,1] range, ensuring consistent magnitudes across dimensions. This property stabilizes dot-product retrieval in RAG and leads to closer alignment with ICL behavior. However, for highly skewed and long-tailed features in the California Housing dataset (e.g., population, income), the majority of samples are compressed into a very narrow interval, while a few outliers dominate the upper bound. This imbalance causes the model to distribute weights unevenly across feature dimensions during training. Our evaluation matrices (prediction difference, cosine similarity, and model difference) further confirm that Min–Max normalization introduces instability on such skewed datasets. In particular, cosine similarity decreases and model difference increases as training progresses, indicating that feature scaling directly impacts the alignment between ICL and RAG dynamics. For additional analyses on Predict Calorie and Wine Quality, please refer to Appendix Figure 6.

## 5 RELATED WORK

Retrieval-augmented generation (RAG) has been extensively studied to enhance language models with external knowledge (Li et al., 2024a; Lewis et al., 2020; Guu et al., 2020; Li & Huang, 2023; Li et al., 2025). Most existing approaches rely on training or fine-tuning both the retriever and generator to effectively integrate retrieved information into downstream tasks. For instance, KIEST (Li & Huang, 2023) dynamically injects entity and attribute knowledge from a knowledge graph during generation, while Li et al. (2025) leveraged feedback from the model's outputs to reward the retriever, thereby improving the relevance of retrieved documents. However, fine-tuning the retriever or predictor requires substantial computational resources. In contrast, in-context learning (ICL) enables models to acquire task-specific behavior from only a few demonstrations, without parameter updates. To better understand this potential, recent research has investigated the underlying mechanism of ICL. Prior

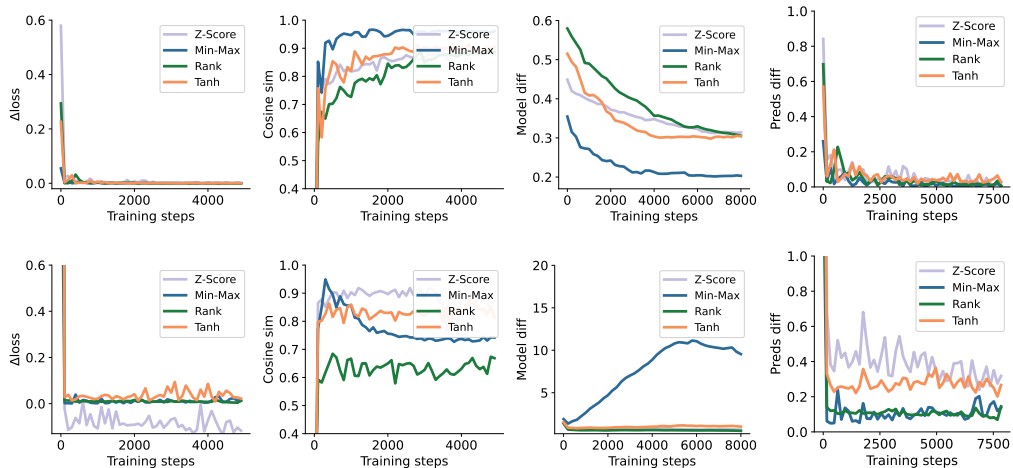

Figure 4: The first row reports the evaluation results for the Bike Sharing dataset across four different normalization methods, using evaluation metrics including the loss difference with the trained Transformer, the training loss of RAG, cosine similarity, model discrepancy, and prediction difference. The second row presents the corresponding results for the California Housing dataset, obtained under the same evaluation protocol.

research has shown that Transformers, particularly linear attention models, can implicitly perform gradient-descent-like updates on in-context data during forward inference Vladymyrov et al. (2024); Zhang et al. (2025b); Von Oswald et al. (2023). Other works have explained the ICL with transformer from the perspective of kernel functions Shen et al. (2025); Ren & Liu (2024), showing that the attention operation can be interpreted as an instance of kernel regression, where queries and keys define feature mappings and the value computation corresponds to regression. Building on these findings, our work takes a step further by investigating whether ICL can reproduce the RAG training. This perspective not only provides theoretical and empirical evidence for the equivalence between ICL and RAG, but also lays the groundwork for accelerating RAG training: by leveraging ICL, models can internalize retrieval-based learning within forward computation, thereby reducing the need for resource-intensive retriever–generator co-training.

## 6 CONCLUSION

Training both the retriever and generator in RAG is often computationally intensive, leading to a trade-off between effectiveness and efficiency. In this paper, we investigate the potential of leveraging in-context learning (ICL) as an alternative mechanism within RAG. We first provided a mathematical perspective on the relationship between in-context learning (ICL) and retrieval-augmented generation (RAG). By constructing an explicit equivalence between linear self-attention Transformers and RAG training under regression tasks, we demonstrated that Transformers trained through gradient descent can effectively simulate RAG behavior. Furthermore, we showed that incorporating deeper layers enables Transformers to refine optimization dynamics. Our empirical analysis highlighted that the distributional properties of real-world datasets critically affect this equivalence, with normalization techniques serving as an effective strategy to stabilize training and improve generalization. Our findings bridge theoretical understanding and empirical evidence, suggesting that ICL can serve as a principled mechanism for optimizing RAG while also motivating future directions in designing retrieval-augmented models that are robust, efficient, and accurate.

## 7 ETHICS STATEMENT

This work is primarily theoretical and empirical in nature, focusing on the connection between in-context learning and retrieval-augmented generation. All datasets used in our experiments are publicly available benchmark datasets (California Housing, Bike Sharing, Wine Quality, and Predict

Calorie Expenditure from Kaggle) that do not contain personally identifiable or sensitive information. Our findings may contribute to more efficient training of retrieval-augmented models, which could reduce computational costs and environmental impact.

## 8 REPRODUCIBILITY STATEMENT

We are committed to ensuring the reproducibility of our results. All datasets used in this work are publicly available: California Housing and Wine Quality from the Kaggle, Bike Sharing from the Kaggle, and Predict Calorie Expenditure from Kaggle. We provide details of the preprocessing steps and dataset splits in Appendix F.

Our models are implemented in JAX and PyTorch, and training configurations (learning rate, batch size, optimizer, number of steps, and model hyperparameters) are documented in the provided training scripts. All experiments were conducted on NVIDIA A100 GPUs.

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

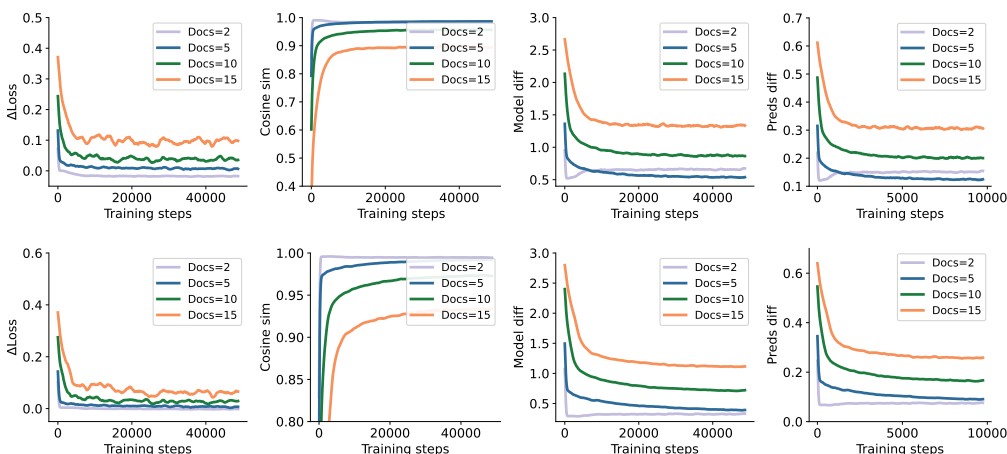

Figure 5: Comparison across layers with varying numbers of documents for the linear projection retriever. The first row corresponds to Layer 2, and the second row corresponds to Layer 5. Each column reports a different evaluation metric: (a) the loss difference between the trained Transformer and RAG, (b) cosine similarity, (c) model difference, and (d) prediction difference.

# A APPENDIX

## A.1 USE OF LLM

Large language models (LLMs) were only used to assist with language polishing and minor grammatical editing of this manuscript.

# B RAG WITH A NON-LINEAR RETRIEVER ACROSS DIFFERENT LAYERS

In the dot-product retriever, the retrieved documents are projected to the same dimensionality as the query and injected into the key–value matrices. This preserves a linear structure in the attention update, making the behavior of RAG closely approximate gradient descent. In contrast, with nonlinear retrievers, the documents are directly concatenated with the input tokens and processed through additional nonlinear layers. This alters the feature space and introduces strong interactions between queries and documents, which accumulate as the number of documents increases, thereby amplifying the discrepancy between ICL and RAG.

Unlike the main analysis with a dot-product retriever, where we reported results for 2, 5, 10, and 25 documents to establish equivalence, in the nonlinear retriever setting we only evaluated 2, 5, 10, and 15 documents. This choice was made because the computational cost grows substantially with larger retrieval sizes, and the divergence from ICL is already evident by 15 documents. We therefore omit 25 document experiments, as the trend is clear without them.

# C MODEL PERFORMANCES ON CALORIE EXPENDITURE AND WINE QUALITY

In the Predict Calorie, we also see the equivalent of the ICL and training of RAG. In the Wine Quality dataset, Min–Max normalization amplifies the influence of outliers, causing most samples to be compressed near zero while a few dominate the scaling. This imbalance not only reduces cosine similarity, as sensitivity vectors diverge from gradient descent, but also increases fluctuations in prediction difference, reflecting instability in the alignment between RAG and ICL dynamics.

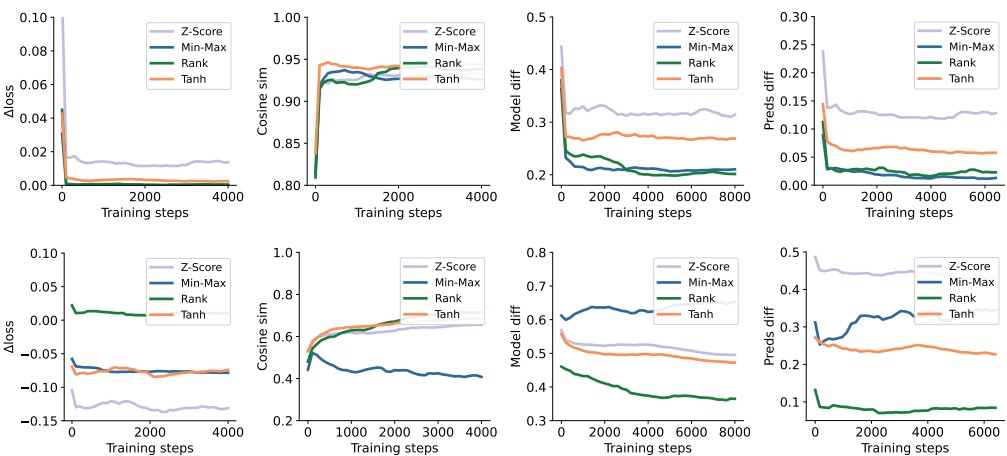

Figure 6: The first row reports the evaluation results for the Predict Calorie Expenditure dataset across four different normalization methods, using evaluation metrics including the loss difference with the trained Transformer, the training loss of RAG, cosine similarity, model discrepancy, and prediction difference. The second row presents the corresponding results for the Wine Quality dataset, obtained under the same evaluation protocol.

## D  RETRIEVER

**Main function**

$$y = (W_q, W_z) \begin{bmatrix} x_q \\ \sum_{i=1}^{n} (W_e x_q)^\top (W_e d_i) \, d_i \end{bmatrix} = W_q x_q + W_z \sum_{i=1}^{n} (W_e x_q)^\top (W_e d_i) \, d_i. \tag{1}$$

**Define $M$ and rewrite the similarity.**

$$M \triangleq W_e^\top W_e \quad \Rightarrow \quad (W_e x_q)^\top (W_e d_i) = x_q^\top W_e^\top W_e d_i = x_q^\top M d_i. \tag{2}$$

Hence,

$$y = W_q x_q + W_z \sum_{i=1}^{n} (x_q^\top M d_i) \, d_i. \tag{3}$$

**Converting "scalar $\times$ vector" into "matrix $\times$ vector."**  Note that $x_q^\top M d_i$ is a scalar, and the following identity holds:

$$(x_q^\top M d_i) \, d_i = d_i (d_i^\top M^\top x_q) = (d_i d_i^\top) M^\top x_q. \tag{4}$$

Therefore,

$$\sum_{i=1}^{n} (x_q^\top M d_i) \, d_i = \sum_{i=1}^{n} (d_i d_i^\top) M^\top x_q = \Big( \sum_{i=1}^{n} d_i d_i^\top \Big) M^\top x_q. \tag{5}$$

**Define the document second-moment matrix $D$.**

$$D \triangleq \sum_{i=1}^{n} d_i d_i^\top \quad \Rightarrow \quad \sum_{i=1}^{n} (x_q^\top M d_i) \, d_i = D M^\top x_q. \tag{6}$$

**Substituting back into $y$.**

$$y = W_q x_q + W_z D M^\top x_q. \tag{7}$$

Then $M = W_e^\top W_e$ is symmetric, i.e., $M^\top = M$. Thus the expression simplifies to

$$y = W_q x_q + W_z D M x_q. \tag{8}$$

The right-hand side is grouped into an equivalent linear mapping:

$$y = (W_q + W_z D M) \, x_q. \tag{9}$$

# E  DETAILS IN PROOF 1

Given a 1-head linear attention layer and tokens $e_j = (x_j^1, x_j^2, y_j)$ for $j = 1, \ldots, N$, we can construct special key, query, and value matrices $W_K, W_Q, W_V$, together with a projection matrix $P$, such that a Transformer update on each token $e_j$ is equivalent to the training progress of the above RAG optimization. More specifically, $e_j \leftarrow (x_1^j, x_2^j, y^j) + (0, -\Delta W_1 x_j, -\Delta W_2 x_j) = (x_1^j, x_2^j, y^j) + PVK^\top q_j$,

In the training of RAG, we model the updated prediction $y'$ as a combination of the original prediction and the change induced by weight updates. Specifically, the difference $y' - y$ reflects how the parameter shifts $\Delta W_1$ and $\Delta W_2$ affect the output through their interaction With input features $x_1$ and $x_2$.

$$y' = W_1' x_1 + W_2' x_2 \tag{10}$$
$$= (W_1 + \Delta W_1) x_1 + (W_2 + \Delta W_2) x_2 \tag{11}$$
$$= W_1 x_1 + \Delta W_1 x_1 + W_2 x_2 + \Delta W_2 x_2 \tag{12}$$
$$= y_1 + \Delta W_1 x_1 + y_2 + \Delta W_2 x_2 \tag{13}$$

The loss function and the one step of gradient descent on L With learning rate $\eta$ yields The Weight change is defined as:

$$L(W_1, W_2) = \frac{1}{2N} \sum_{i=1}^N \left( W_1 x_1^i + W_2 x_2^i - y_i \right)^2 \tag{14}$$

$$\Delta W_1 = -\eta \nabla_{W1} L(W1) = -\frac{\eta}{N} \sum_{i=1}^N \left( W_1 x_1^i + W_2 x_2^i - y_i \right) \cdot x_1^i \tag{15}$$

$$\Delta W_2 = -\eta \nabla_{W2} L(W2) = -\frac{\eta}{N} \sum_{i=1}^N \left( W_1 x_1^i + W_2 x_2^i - y_i \right) \cdot x_2^i \tag{16}$$

$$\Delta y = \Delta y_1 + \Delta y_2 = \Delta W_1 x_1 + \Delta W_2 x_2 \tag{17}$$
$$\tag{18}$$

Define y:

$$\Delta y = \left( -\frac{\eta}{N} \sum_{i=1}^N (W_1 x_1^i + W_2 x_2^i - y_i) x_1^i \right) x_1^j + \left( -\frac{\eta}{N} \sum_{i=1}^N (W_1 x_1^i + W_2 x_2^i - y_i) x_2^i \right) x_2^j \tag{19}$$

$$\Delta y = -\left( \left( \frac{\eta}{N} \sum_{i=1}^N (W_1 x_1^i + W_2 x_2^i - y_i) x_1^i \right) x_1^j + \left( \frac{\eta}{N} \sum_{i=1}^N (W_1 x_1^i + W_2 x_2^i - y_i) x_2^i \right) x_2^j \right) \tag{20}$$

$$-\Delta y = \left( \left( \frac{\eta}{N} \sum_{i=1}^N (W_1 x_1^i + W_2 x_2^i - y_i) x_1^i \right) x_1^j + \left( \frac{\eta}{N} \sum_{i=1}^N (W_1 x_1^i + W_2 x_2^i - y_i) x_2^i \right) x_2^j \right) \tag{21}$$

The update of the target is denoted as:

$$\begin{pmatrix} x_1^j \\ x_2^j \\ y^j \end{pmatrix} \leftarrow \begin{pmatrix} x_1^j \\ x_2^j \\ y^j \end{pmatrix} + \begin{pmatrix} 0 \\ 0 \\ -\Delta y \end{pmatrix} \tag{22}$$

$$\begin{pmatrix} 0 \\ 0 \\ -\Delta y \end{pmatrix} = \begin{pmatrix} 0 \\ 0 \\ -(\Delta W_1 x_1 + \Delta W_2 x_2) \end{pmatrix} \tag{23}$$

$$\begin{pmatrix} 0 \\ 0 \\ -\Delta y \end{pmatrix} = \frac{\eta}{N} \sum_{i=1}^{N} \begin{pmatrix} 0 & 0 & 0 \\ 0 & 0 & 0 \\ (W_1 x_1^i + W_2 x_2^i - y^i) x_1^i & (W_1 x_1^i + W_2 x_2^i - y^i) x_2^i & 0 \end{pmatrix} \begin{pmatrix} x_1^j \\ x_2^j \\ 0 \end{pmatrix} \quad (24)$$

$$= \frac{\eta}{N} \sum_{i=1}^{N} \begin{pmatrix} 0 \\ 0 \\ (W_1 x_1^i + W_2 x_2^i - y^i) \end{pmatrix} \otimes \begin{pmatrix} x_1^i & x_2^i & 0 \end{pmatrix} \begin{pmatrix} x_1^j \\ x_2^j \\ 0 \end{pmatrix} \quad (25)$$

$$= \frac{\eta}{N} \sum_{i=1}^{N} \left( \begin{pmatrix} 0 & 0 & 0 \\ 0 & 0 & 0 \\ W1 & W2 & -1 \end{pmatrix} \begin{pmatrix} x_1^i \\ x_2^i \\ y \end{pmatrix} \right) \otimes \begin{pmatrix} x_1^i & x_2^i & 0 \end{pmatrix} \begin{pmatrix} x_1^j \\ x_2^j \\ 0 \end{pmatrix} \quad (26)$$

$$= \frac{\eta}{N} \sum_{i=1}^{N} \left( \begin{pmatrix} 0 & 0 & 0 \\ 0 & 0 & 0 \\ W1 & W2 & -1 \end{pmatrix} \begin{pmatrix} x_1^i \\ x_2^i \\ y \end{pmatrix} \right) \otimes \begin{pmatrix} x_1^i & x_2^i & 0 \end{pmatrix} \begin{pmatrix} 1 & 0 & 0 \\ 0 & 1 & 0 \\ 0 & 0 & 0 \end{pmatrix} \begin{pmatrix} x_1^j \\ x_2^j \\ y \end{pmatrix} \quad (27)$$

$$= \frac{\eta}{N} \sum_{i=1}^{N} \left( \begin{pmatrix} 0 & 0 & 0 \\ 0 & 0 & 0 \\ W1 & W2 & -1 \end{pmatrix} \begin{pmatrix} x_1^i \\ x_2^i \\ y \end{pmatrix} \right) \otimes \left( \begin{pmatrix} 1 & 0 & 0 \\ 0 & 1 & 0 \\ 0 & 0 & 0 \end{pmatrix} \begin{pmatrix} x_1^j \\ x_2^j \\ y \end{pmatrix} \right)^{\top} \left( \begin{pmatrix} 1 & 0 & 0 \\ 0 & 1 & 0 \\ 0 & 0 & 0 \end{pmatrix} \begin{pmatrix} x_1^j \\ x_2^j \\ y \end{pmatrix} \right)$$
$$(28)$$

So, the right part in equation 28 is equal to the right part in equation 23.

## F  THE DETAILS OF THE DATASET

- **California Housing**: Given eight features — ['MedInc', 'HouseAge', 'AveRooms', 'AveBedrms', 'Population', 'AveOccup', 'Latitude', 'Longitude'] — the task is to predict MedHouseVal. The dataset is split into 16,640 training samples and 2,000 test samples.

- **Bike Sharing**: Using the features ['season', 'yr', 'mnth', 'hr', 'holiday', 'weekday', 'workingday', 'weathersit', 'temp', 'atemp', 'hum', 'windspeed', 'casual', 'registered'], the task is to predict count. The dataset contains 15,641 training samples and 1,738 test samples.

- **Wine Quality**: Given eleven physicochemical features — [fixed acidity, volatile acidity, citric acid, residual sugar, chlorides, free sulfur dioxide, total sulfur dioxide, density, pH, sulphates, alcohol] the task is to predict the wine quality (a sensory score ranging from 0 to 10). The dataset is split into 4,408 training samples and 490 test samples.

- **Predict Calorie Expenditure**: Using the features [Gender, Age, Height, Weight, Duration, Heart_Rate, Body_Temp], the task is to predict the number of Calories expended. The dataset is split into 13,500 training samples and 1,540 test samples.

