# OpenReview forum: "Retrieval-Augmented Generation as In-Context Optimization: A Gradient Descent Perspective"
_ICLR.cc/2026/Conference — ICLR 2026 Conference Withdrawn Submission_

### Official Review · Reviewer_Qzcg · 2025-10-30

**Soundness:** 3
**Presentation:** 3
**Contribution:** 2
**Rating:** 4
**Confidence:** 4

**Summary:**

The paper argues that trained Transformers can perform RAG via in-context optimization. It proves, in a linear self-attention setup, an explicit weight construction where a self-attention layer’s data transformation is equivalent to one gradient-descent step of a simplified, jointly trained RAG on a regression loss; stacking layers approximates multi-step updates with curvature correction. Empirically, self-attention-only Transformers mimic gradient-trained RAG on simple regression tasks, and the authors show that data distribution strongly limits generalization in nonlinear settings, where normalization improves stability and out-of-distribution performance.

**Strengths:**

Strengths

- Clear theory: gives an explicit weight construction showing that a self-attention layer can emulate one gradient step of a simplified, trainable RAG under MSE.

- Unifying view: connects RAG training and in-context learning under one linear framework, which clarifies “why ICL can look like optimization.”

- Empirical support: reproduces the predicted behavior on controlled regression tasks and tracks multi-step dynamics with deeper attention.

- Practical insight: shows that normalization can improve out-of-distribution stability for ICL-style setups.

**Weaknesses:**

Weaknesses

- Narrow assumptions: main guarantees require linear self-attention and squared-loss regression; the bridge to realistic, non-linear NLP tasks is indirect.

- Limited benchmarks: experiments focus on synthetic or simplified settings rather than strong real-world RAG workloads.

- Retrieval simplifications: the retriever is linearized; it is unclear how results transfer to modern dense or hybrid retrievers.

- Partial equivalence: the construction matches one or a few optimization steps; it does not prove full training equivalence for deep, non-linear models

**Questions:**

same as the weaknesses mentioned above.

---

### Official Review · Reviewer_MhfR · 2025-10-31

**Soundness:** 3
**Presentation:** 3
**Contribution:** 3
**Rating:** 4
**Confidence:** 4

**Summary:**

This paper explores the theoretical connection between Retrieval-Augmented Generation (RAG) and In-Context Learning (ICL) from a gradient-descent perspective. The authors formalize a linearized RAG model and show that the update step in this model can be viewed as equivalent to one step of gradient descent, which in turn corresponds to the forward computation of a linear self-attention layer. The paper is well written and the mathematical derivations are generally clear. However, the novelty and conceptual contribution appear limited, as the core claim largely reiterates the now well-established view that ICL performs implicit gradient-based optimization.

**Strengths:**

1. The paper presents a clean and mathematically rigorous derivation connecting RAG-style architectures and gradient descent dynamics.

2. The experiments are well-structured and provide numerical evidence that linear attention layers can indeed approximate explicit gradient steps.

3. The work offers a pedagogically useful framing that may help unify perspectives between retrieval-based models and in-context learning.

4. The writing is clear and the figures (e.g., linear vs nonlinear settings) are helpful for understanding.

**Weaknesses:**

1. The central message that “ICL is equivalent to performing gradient descent” has already been explored in several prior works（https://arxiv.org/abs/2212.10559） . This paper essentially restates that idea in the context of RAG, without introducing new theoretical mechanisms or insights beyond this known equivalence.

2. The paper treats RAG as if it involves an explicit gradient-based optimization over retriever and generator components. In practice, RAG does not require such training to function — retrieval is typically non-parametric, and the generator is often pretrained and frozen. Therefore, the “RAG gradient descent” described in the paper is a synthetic construct, not a realistic depiction of how RAG systems are optimized. As such, the equivalence shown is somewhat tautological: it effectively re-derives the “ICL = gradient descent” result under a contrived RAG formulation.

3. The theoretical model replaces discrete retrieval (top-k search) with continuous linear projection, removing the core difficulty of retrieval-augmented systems. Consequently, the work sidesteps what truly differentiates RAG from standard attention mechanisms — the non-differentiable retrieval and external memory integration.

4. All experiments are conducted on synthetic or low-dimensional regression tasks. There is no evaluation on real RAG settings such as question answering or document-grounded generation, which limits the empirical relevance.

5.While the theory is elegant, it remains unclear how this connection could guide the design or training of actual retrieval-augmented models. The paper stops short of suggesting how such an equivalence could improve efficiency, generalization, or robustness in realistic scenarios.

**Questions:**

none

---

### Official Review · Reviewer_ZyM9 · 2025-10-31

**Soundness:** 2
**Presentation:** 2
**Contribution:** 1
**Rating:** 2
**Confidence:** 3

**Summary:**

This paper investigates the question of whether in-context learning (ICL) can serve as an alternative mechanism for retrieval-augmented generation (RAG). The authors claim positive by showing one-layer linear self-attention can implement the same operation as training a simplified RAG system for joint document selection and output prediction. The authors then test this claim on more realistic transformers and regression datasets and found distributional properties of real-world datasets affect this equivalence. Normalization techniques are proposed to mitigate the issue.

**Strengths:**

This paper attends to an interesting question of how ICL can be an alternative to RAG.

**Weaknesses:**

- On the theoretical side, it's unclear how establishing an equivalence between a simplified attention layer and a simplified RAG system helps address the target question of this paper. The equivalence appears to arise mainly from the particular simplification choices for ICL and RAG, rather than reflecting a deeper underlying link between them.

- On the experimental side, the proposed normalization technique appears somewhat brittle, i.e., unstable on skewed datasets. This suggests that using ICL for RAG still requires careful, data-dependent tuning, which undermines some of its efficiency benefits.

- I found the notations in Section 2 difficult to follow. Below are some concrete points of confusion.

  It'd be helpful to define the dimensionality of the symbols in Section 2.1, as was done in Section 2.2.

  The symbol $x$ is used inconsistently. The query input is sometimes denoted as $x_q$ and sometimes as $x^q$. The use of subscripts and superscripts also varies. For example, $x_i^q$ and $x^i_1$ appear, but it's unclear how these are related (or unrelated).

  I'm having trouble understanding $x=W_e(x),d_i=W_e(d_i)$ in line 152. It’s not obvious whether this is meant as code or a mathematical equation.

  The notation $\otimes$ in Equation (8) seems to be undefined.

**Questions:**

In line 254, "feather size" appears to be a typo.

---

### Note · Authors · 2026-01-04

I have read and agree with the venue's withdrawal policy on behalf of myself and my co-authors.